# Unfiltered: Measuring Cloud-based Email Filtering Bypasses

## ABSTRACT

Email service has increasingly been outsourced to cloud-based providers and so too has the task of filtering such messages for potential threats. Thus, customers will commonly direct that their incoming email is first sent to a third-party email filtering provider (e.g., Proofpoint or Barracuda) and only the "clean" messages are then sent on to their email service provider (e.g., Gmail or Microsoft Exchange Online). However, this loosely coupled approach can, in theory, be *bypassed* if the email service provider is not configured to *only* accept messages that arrive from the email filtering service. In this paper we demonstrate that such bypasses are commonly possible. We document a multi-step methodology to infer if an organization has correctly configured its email service provider to guard against such scenarios. Then, using an empirical measurement of *edu* and *com* domains as a case study, we show that 80% of such organizations making use of popular cloud-based email filtering services can be bypassed in this manner. We end by discussing potential reasons why such misconfigurations can occur and outlining the complexities and challenges in hardening the binding between email filtering providers and email service providers.

## 1 INTRODUCTION

Over the last decade, a range of economic incentives have driven enterprises to abandon key self-hosted services and outsource these functions to third-party cloud-based service providers. This trend has encompassed services including storage (e.g., Dropbox, GDrive, Skydrive, etc.), backup (e.g., Backblaze), domain names (e.g., Amazon Route 53, Cloudflare DNS), productivity applications (e.g., GSuite, Microsoft 365), web hosting (e.g., Cloudflare, Akamai, AWS) and, importantly for this paper, email (e.g. Gmail, and Microsoft Exchange Online). However, there is no established or standardized implementation or protocol for composing such cloud services, and thus each situation is managed in an application-specific ad hoc manner. In this paper we focus on a simple example of this situation — the interaction between cloud-based email services and email filtering services.

While third-party email services commonly used by enterprises (i.e., Gmail or Microsoft Exchange Online) provide native email filtering capabilities, many organizations prefer to supplement these capabilities with specialized third-party filtering services, such as those offered by Proofpoint [49] or Barracuda [3].[1] As with their on-premises appliance predecessors, such services offer greatly enhanced policy control and configurability, more advanced security features (e.g., URL-rewriting and attachment "detonation"), rich reporting capabilities and market themselves as being singularly focused on defending against the latest email-borne threats.

Implementing this filtering step in the cloud requires a mechanism to manage the flow of inbound email — funneling these *first* to the filtering service and *then* to the email service. While there are a range of ways such a capability could be implemented in principle,

in practice the common mechanism is to configure a domain's DNS MX record to direct incoming traffic to the email filtering provider and then configure the email filtering provider to deliver the filtered email stream on to the domain's email service provider. However, while this procedure ensures that filtered email is ultimately delivered, it does not guarantee that delivered email has been filtered. Indeed, a clever adversary might determine the server used by a domain's email service provider and send malicious mail *directly* to them — thus bypassing the third-party filtering (and the security benefits it provides). There are a number of ad hoc measures that an enterprise might take to protect against such actions (e.g., rejecting email from IP addresses not operated by the email filtering service) but such defensive configurations are neither required for correct operation nor are they externally visible to any outside auditor.

This paper investigates the deployment of cloud-based email filtering services via two primary contributions:

- Through careful controlled trials, we have developed and validated a multi-step measurement procedure to infer a domain owner's choice of email service provider, email filtering service provider, and the integrity of the binding between the two (i.e., whether the filtering relationship is "bypassable").
- Using this technique, we have conducted case studies focused on auditing 673 edu domains and a sample of 928 popular com domains using the 15 most-commonly seen cloud-based email filtering services (e.g., Proofpoint, Barracuda, Cisco, Mimecast). Of these, we show that 80% of these domains do not protect the integrity of the mail delivery path and therefore their filtering can be trivially bypassed.

Finally, based on this experience, we describe the challenges and tradeoffs involved in addressing this problem — itself a special case of an overall challenge in architecting composition between third-party cloud services.

## 2 BACKGROUND

We begin by reviewing key terminology and protocols involved in email transmission (SMTP), to provide background context for specific steps in our measurement methodology (§ 3). Next, we provide a high-level overview for how email delivery works in the presence of a cloud-based email filtering service, and describe how a clever attacker might bypass such a setup without appropriate countermeasures. Subsequently, we discuss how email service providers can be properly configured to mitigate such bypassing threats.

### 2.1 Simple Mail Transfer Protocol

The simple mail transfer protocol (SMTP) is a family of protocols that governs the transmission of email messages [25], including email forwarding and delivery. All protocols in the SMTP family are text-based and follow a similar session-based model [30]. Figure 1 depicts a typical SMTP session between a client (C) and a server

---

[1]Liu et al's 2021 study of email providers shows that 13% of the Alexa Top 1k domains made use of one of these two services for this purpose [30].

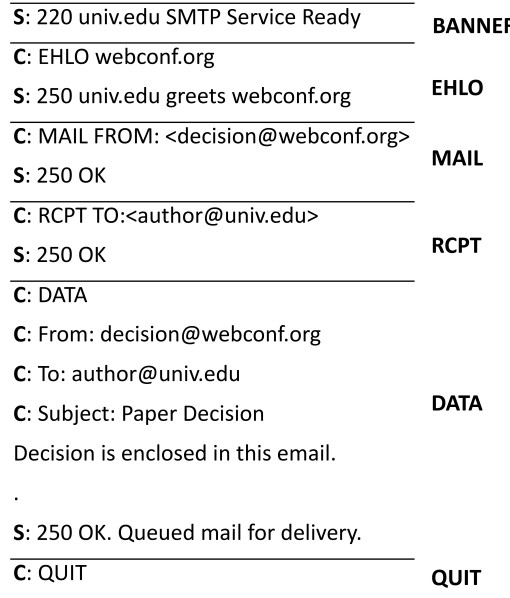

| | |
|---|---|
| **S**: 220 univ.edu SMTP Service Ready | **BANNER** |
| **C**: EHLO webconf.org | |
| **S**: 250 univ.edu greets webconf.org | **EHLO** |
| **C**: MAIL FROM: <decision@webconf.org> | |
| **S**: 250 OK | **MAIL** |
| **C**: RCPT TO:<author@univ.edu> | |
| **S**: 250 OK | **RCPT** |
| **C**: DATA | |
| **C**: From: decision@webconf.org | |
| **C**: To: author@univ.edu | |
| **C**: Subject: Paper Decision | **DATA** |
| Decision is enclosed in this email. | |
| . | |
| **S**: 250 OK. Queued mail for delivery. | |
| **C**: QUIT | **QUIT** |

**Figure 1: A typical SMTP session between a client (C) and a server (S), which handles mail for univ.edu.**

(S). The SMTP session starts when the client initiates a successful connection with the server. After establishing this connection, the client and the server exchange information with commands and responses. They start by announcing each other's identity in BANNER and EHLO messages. Next, the client specifies the email address of the sender (in this case, decision@webconf.org) within the MAIL command. The server responds with a 250 message code if it does not encounter any problems. The client then specifies the email address and information about the recipient of the email (i.e., author@univ.edu) in the RCPT command (hence referred to as the RCPT address). The server again acknowledges the command with a 250 message code if no issues are encountered, otherwise returning a *specific error code*. Next, the client sends the contents of the email (starting with a DATA command), and ending with a period ("."). After receiving this data from the client, the server acknowledges the command and delivers the email to the recipient's mailbox. While the example depicted in Figure 1 represents a traditional case in which the server is hosted by univ.edu, it is increasingly common that this mail service is instead outsourced to Gmail or Microsoft Exchange Online.

### 2.2 Email Delivery with Filtering Services

Organizations have increasingly adopted cloud-based email filtering services to defend against various email-based threats [30]. These cloud services act as gateways between the Internet and organizations' email servers, expanding the process of email delivery beyond one simple SMTP session, as illustrated in Figure 2. First, the sender uses their Mail User Agent (MUA) to craft and submit an email to their email server (step 1).[2] The sender's email server

---

[2]In context of Web-hosted third-party mail services, such as provided by Gmail, the MUA and email server may in fact be part of the same service offering.

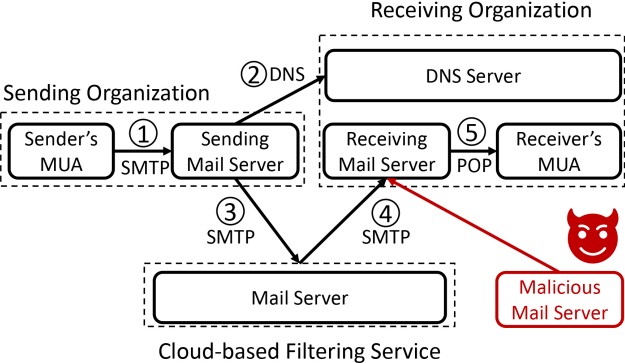

**Figure 2: The email transmission flow when a recipient's organization uses a cloud-based filtering service, and how an attacker can bypass the filtering service for insecurely configured organizations.**

then identifies the recipient's email server by querying the DNS MX record associated with the recipient's domain (step 2). If the recipient's organization uses a cloud-based email filtering service, the recipient's MX record points to an email server hosted by the filtering service. The sender's email server then initiates an SMTP session with the filtering service's email server and sends the email to them (step 3). After processing the email (e.g., spam filtering and URL rewriting), the filtering service then forwards the email to the recipient's mail server (step 4). Subsequently, the recipient can then retrieve the email from their organization's mail server and display the message using their MUA (step 5).

### 2.3 Bypassing Email Filtering Services

The above setup works when the sender follows the normal email transmission flow (i.e., looking up the recipient's MX record and then sending their email to the mail server designated by the MX record). However, as mentioned earlier, a clever adversary might bypass the filtering service by directly sending an email to the recipient's email server ("direct delivery"), once they infer the recipient's email service provider. Figure 2 depicts an example of this attack. Instead of taking step 2, 3, and 4, the adversary directly delivers email to the recipient's mail server, bypassing the filtering service and any protection provided by it. Analyzing the extent to which organizations using cloud-based filtering services are susceptible to these kinds of bypass attacks is the focus of our work.

To prevent such bypasses from happening, organizations can harden the binding between their email filtering service and their email server by configuring their email server to only accept messages from the filtering service's servers.

### 2.4 Configuring Email Servers to Only Accept from Filtering Service

By comprehensively surveying the documentation of major email filtering services and major email services, we have identified three email service providers that allow for a mechanism to appropriately restricting inbound email delivery for this purpose: Gmail – the email server for Google Workspace customers, Microsoft Exchange

Online (hence referred to as "Exchange Online") – the email server for Microsoft 365 customers, and Zoho Mail (hence referred to as "Zoho"). All three email services share the same underlying idea: provide an "allow list" of the IP addresses used by a domain's filtering service and only accept inbound email destined for the domain from IP addresses on that list.

While the underlying idea is the same across all three providers, the configuration syntax varies. For Gmail and Zoho, organizations first specify the IP addresses of their filtering service's servers in a dedicated "inbound gateway IP list", and then enable a separate feature that rejects all email not from gateway IPs. By contrast, Exchange Online, does not have a dedicated configuration option for this purpose. Instead, organizations can specify the filtering services' IP addresses and associate rules using a "connector" [35], which is a general tool for customizing email routing [4, 42, 54].[3]

## 3 METHODOLOGY

While the *potential* for such email filter bypassing is evident in the design and documentation of these services, whether such vulnerabilities exist in practice is a separate empirical question. To explore this further, we must first identify those organizations using third-party mail filtering services, determine the underlying mail service provider to which their filtered mail will be delivered, and then establish, via measurement or inference, the integrity of their delivery path (i.e., whether such a bypass is feasible).

### 3.1 Identifying Mail Filtering Service

As described earlier, third-party mail filtering services are designed to be the first point of contact in a domain's mail delivery path. This is achieved by setting a domain's MX record to direct all inbound messages to filtering service. However, this mechanism can be implemented in a number of ways in practice. For example, foo.com might set its MX record to xxx.gslb.pphosted.com (a domain operated by Proofpoint), or it might point to inbound.foo.com where inbound.foo.com's A record is set to resolve to an IP address in the prefix 147.163.128.0/19 (a prefix operated by Proofpoint) or even more complex combinations using CNAME and multiple levels of name resolution or proxying.

In prior work focused on identifying mail service providers, Liu et al. [30] document how the combination of MX record, A record, TLS certificate (for domains accepting TLS for SMTP mail delivery), SMTP banner and protocol response can be combined to obtain high confidence assessment of the organization accepting mail delivery for a domain. Using a variant of this approach we develop "signatures" for a set of fifteen leading mail filtering services: Proofpoint, Mimecast, Cisco (aka Ironport), Barracuda, TrendMicro, Broadcom (aka Symantec), FireEye (aka Trellis), Sophos, Cloudflare, Fortinet, Solarwinds, Forcepoint, AppRiver, Spamhero and HornetSecurity.[4]

We then apply this approach to corpus of registered edu and com domains. Since EDUCAUSE (the registry for edu) does not publish its DNS zone files, we construct this list using edu containing X.509 certificates collected by Censys [11] using mass scanning. For

our com set, we use the 50k most popular domains as identified by Google's Chrome User Experience Report (CrUX) [17]. We remove any domains that do not have valid MX records (i.e., do not accept mail) and further extract the subset that make use of mail filtering services (using the signatures we described previously). These steps produce a corpus of 889 edu domains and 1,429 com domains that make use of one of these 15 services (15–17% of each corpus). Consistent with prior findings, Proofpoint is the dominant service provider in our data, followed by Barracuda, Mimecast and Cisco which together serve 89% of the domains using such third-party mail filtering.

### 3.2 Inferring Mail Service Provider

While a domain's use of a mail filtering provider can be measured directly, *where* such a provider subsequently delivers the filtered mail is not directly visible. Since few domains make public their choice of mail service provider, we have developed measurement workflows to infer with high confidence if a filtered domain uses Google, Microsoft or Zoho as their backend email service provider.[5] By analyzing the documentation of these three providers, along with insights gleaned from mail administrator postings, and by empirically creating and testing our own subscriptions to these services, we establish that all three providers expose some externally visible state when an organization has a valid subscription (and, crucially, this state is not evident when the subscription is deleted or defunct).

In particular, it is documented that when an organization has a valid subscription with Gmail, Google automatically creates a postmaster and an abuse email address associated with the organization's domain name [16, 18, 64]. Similarly, Zoho also creates a default postmaster and abuse address for each domain that has a valid subscription [32]. While Exchange Online does not automatically create any default email addresses for an organization, it automatically creates a uniquely formatted MX record under the mail.protection.outlook.com subdomain [2, 36, 38]. For example, if the domain univ.edu has contracted for service with Exchange Online, then Microsoft will create an MX record named univ-edu.mail.protection.outlook.com [38, 44].[6]

Critically, each of these pieces of state is externally testable. Thus, we can infer whether a domain foo.com is associated with a valid Gmail subscription by connecting to Gmail's SMTP servers and specifying the postmaster@foo.com address as a parameter to the RCPT command. If foo.com is not hosted by Gmail then the service will return a 550 error code, otherwise it will return an OK or more specific error depending on configuration (as we will discuss later). Similarly, Zoho's SMTP server will return a 553 Relaying disallowed error code if the domain does not have a current subscription to Zoho, and 250 OK response code otherwise.

---

[3]The generality of this mechanism can be confusing and, perhaps as a result, we observe that some filtering service documentation incorrectly directs their customers to implement insecure configurations.

[4]This list captures the leading mail security providers in two industry reports on the sector [14, 59] as well as a few others that appeared non-trivial times in our data.

[5]We select these three because they are the major mail service providers that provide a mechanism to secure the mail delivery path; absent such a mechanism, all other mail service providers are *de facto* "bypassable". Moreover, as identified in previous work, Google's Gmail and Microsoft's Exchange Online dominate the third-party mail service provider market — implementing the mail backend of roughly 40–45% of well-trafficked com domains [30].

[6]Also, as per Microsoft's documentation, organizations can optionally create a CNAME record that enables certainly mail clients (like Outlook) to automatically discover the Exchange server used and configure themselves correctly [34]. This CNAME record can similarly be used to infer the use of Exchange Online as an email service, as prior studies have done [31].

| | C: RCPT TO: <postmaster@univ.edu> | |
|---|---|---|
| **GMAIL** | **S**: 550 5.1.1 The email account that you tried to reach does not exist. | **RCPT** |

| | C: RCPT TO: <postmaster@univ.edu> | |
|---|---|---|
| **ZOHO** | **S**: 553 Relaying disallowed | **RCPT** |

**Figure 3: Gmail returns a 550 error code, and Zoho returns a 553 error code when the recipient address does not exist.**

Figure 3 illustrates this behavior (for Gmail and Zoho respectively). Note that none of these tests require completing an SMTP transaction and thus do not involve sending messages to the associated accounts. For Exchange Online, we can infer that a domain is associated with a valid subscription to the service if the appropriate DNS record exists (e.g., `univ-edu.mail.protection.outlook.com` for the domain `univ.edu`).

Finally, while this inference technique is both easy to perform and accurate, it is unable to distinguish between an organization that makes active use of a mail service (e.g., Gmail) from one which merely has an active subscription (e.g., an organization that uses Google cloud storage, and hence has a Google Workspace subscription, but uses another provider for email). To avoid implicit bias from this effect, we adopt a conservative approach to limit the set of domains we consider to those showing evidence of active email use. For this step we rely on the Sender Policy Framework (SPF), a widely-deployed email protocol designed to help prevent attackers from spoofing email. As part of this protocol, a domain publishes a DNS TXT record specifying the list of domains and IP addresses authorized to send email on its behalf [24]. Then, when actively using a third-party email service provider, organizations typically list the IP addresses of their provider in the SPF record (otherwise mail sent via the provider will be rejected or sent to spam folders by many recipients). We use Izhikevich et al.'s ZDNS [23] to parse each domain's SPF record, recursive querying to expand any "included" entries to handle common SPF configuration styles (these details described more fully in Appendix A).

After this filtering, 673 edu domains and 928 com domains remain — those actively making use of one of the three email service providers *and* using one of the 15 mail filtering services (see Table 1).[7] It is this set of domains that we test for "bypassability".

### 3.3 Inferring Mail Path Integrity

As discussed, the core concern of this paper is to understand whether a third-party mail filtering service can be bypassed by sending directly to a domain's backend mail service provider. Ultimately, this question is determined entirely by the configuration of the backend mail service — whether it will accept inbound email for a domain from any party, or if it will only accept such messages from the domain's mail filtering service.

| | C: RCPT TO: <postmaster@univ.edu> | |
|---|---|---|
| **GMAIL** | **S**: 421 4.7.0 IP not in whitelist for RCPT domain, closing connection. | **RCPT** |

| | C: DATA | |
|---|---|---|
| | C: From: <decision@webconf.org> | |
| | C: To: <postmaster@univ.edu> | |
| | C: Subject: Paper Decision | |
| **EXCHANGE** | Decision is enclosed in this email. | **DATA** |
| | . | |
| | **S**: 550 5.7.51 TenantInboundAttribution; There is a partner connector configured that matched the message's recipient domain. | |

**Figure 4: Gmail returns a 421 error with a correctly configured protective "gateway", and Exchange Online returns a 550 error with a correctly setup protective "connector".**

This configuration behavior is revealed when an unauthorized party (i.e., from an IP address not belonging to the mail filtering service) initiates an SMTP transaction with the mail service and attempts to address mail to *valid addresses* in the domain. Based on systematic empirical testing, we have determined that securely-configured domains hosted by Gmail and Zoho will reject such an email during the RCPT stage of a session, while Exchange Online will reject during the DATA stage. Figure 4 illustrates this behavior (for Gmail and Exchange Online respectively). Here, messages are addressed to valid accounts in the domain, but the mail service provider is configured to only accept inbound mail from its mail filtering service. Here, Gmail returns an error code 421 with the message "IP not in whitelist for RCPT domain" (in response to the RCPT command) while Exchange Online returns error code 550 with the message "There is a partner connector configured that matched the message's recipient domain" (in response to the DATA command). Conversely, if the domain owner has not configured such inbound mail restrictions, then standard "250 OK" responses will be returned. A similar test distinguishes securely-configured Zoho-hosted domains. By connecting with each domain's back-end mail servers and conducting such integrity tests we can infer whether their mail delivery path is secure or if it can be bypassed.[8]

For Gmail and Zoho, this integrity test is trivial to perform since, by default, there are well-known valid addresses (i.e., postmaster and abuse) for each hosted domain. However, Exchange Online has no such defaults and thus for this service we are forced to rely on a heuristic. Complicating this further is the fact that Exchange Online's integrity test takes place via the DATA command. Hence, if it succeeds, a message will be delivered to the recipient. Since we wish to avoid imposing any undue burden on regular users of such email services, we cannot simply probe using common names (e.g., alice or bob) or addresses obtained via search engines.

---

[7]The distribution of mail filtering services in this conservatively filtered set is consistent with the same distribution in the original corpus, suggesting that there is no correlation between the active use of the service and the choice of mail filtering service provider.

[8]Note that Gmail and Zoho SMTP servers use well-known DNS addresses that accept mail traffic for all customers, while domains hosted by Exchange Online receive inbound mail via a unique domain-specific address as described earlier.

Instead, we first note that Exchange Online has a per-domain option, Directory-Based Edge Blocking [41], which causes the service to reject invalid addresses up front. However, if this feature is not enabled, then the integrity test can be performed using invalid addresses. Thus, we probe Exchange Online-hosted domains using a 25 character randomly generated alphanumeric email address in the RCPT command. If this address is accepted, then the invalid address blocking feature is disabled and this address will serve as a *valid address* for the purpose of integrity testing. If not, we then resort to blindly probing using a set of well-known *administrative addresses* (e.g., postmaster, admin, info, etc) as identified by Bennett et al. [6]. If any of these addresses are accepted in the RCPT command we then proceed with the previously described integrity test. Otherwise, we record the integrity of the domain as unknown. Only in one particular case — an Exchange Online-hosted domain blocking invalid addresses has configured one of the well-known administrative email addresses *and* has not correctly configured the mail service to limit the bypass of the mail filtering service — will we end up delivering an email (we discuss this further in Section 5).

## 3.4 Limitations

Our methodology has several limitations. In particular, it is based on assumptions that are well-suited to standard modes of use, but may fail — either inducing false positives or false negatives — with certain edge cases.

First, the presence or absence of a email service provider's domain or IP addresses in a domain's SPF record is not an *air tight* indicator of whether the organization is currently using the email service. For example, an organization might have migrated to another email provider and not yet updated their DNS record – thus mischaracterizing a site as vulnerable when it may not be (because they are not using that mail service provider). Similarly, an organization might be a Gmail customer yet decide to route all outbound email through its email filtering service (and thus only include the filtering service in their SPF record). Such sites would be ignored by our analysis even though they may be vulnerable because we cannot determine their mail service provider. While we believe such situations are atypical today, that might change in the future.

There are similarly rare edge cases around the implementation of mail service provider inbound mail filtering. While our methodology is focused around inferring the use of recommended best practice (as found in both mail and filtering service provider documentation) we have seen ad-hoc configurations that attempt to achieve the same effect (e.g., using Exchange Online's transport rules [37] to silently filter inbound email). In such cases, we might mistakenly decide that a site's mail filtering service is bypassable, while such an ad hoc filtering solution in fact protects it.

Finally, it is possible to use mail filtering services with self-hosted email and these, as well, may not secure the mail delivery path from bypass. However, they are outside the scope of our current approach and will not be captured by our methodology.

## 4 RESULTS

We identify 673 edu and 928 com domains that use notable third-party filtering services and email providers. In this section we show the results of our various inference techniques for these

| Filtering Service | Domains | | edu | | com | |
|---|---|---|---|---|---|---|
| Proofpoint | 720 | (45%) | 213 | (32%) | 507 | (55%) |
| Barracuda | 283 | (18%) | 245 | (36%) | 38 | (4%) |
| Mimecast | 254 | (16%) | 69 | (10%) | 185 | (20%) |
| Cisco | 160 | (10%) | 96 | (14%) | 64 | (7%) |
| TrendMicro | 43 | (3%) | 8 | (1%) | 35 | (4%) |
| Sophos | 29 | (2%) | 17 | (3%) | 12 | (1%) |
| FireEye | 18 | (1%) | | – | 18 | (2%) |
| Cloudflare | 18 | (1%) | 1 | (0.1%) | 17 | (2%) |
| AppRiver | 18 | (1%) | 8 | (1%) | 10 | (1%) |
| Broadcom | 15 | (0.9%) | 2 | (0.3%) | 13 | (1%) |
| ForcePoint | 14 | (0.8%) | 3 | (0.4%) | 11 | (1%) |
| Fortinet | 14 | (0.8%) | 5 | (0.7%) | 9 | (0.9%) |
| Hornetsecurity | 8 | (0.5%) | 2 | (0.3%) | 6 | (0.6%) |
| Solarwinds | 4 | (0.2%) | 2 | (0.3%) | 2 | (0.2%) |
| Spamhero | 3 | (0.2%) | 2 | (0.3%) | 1 | (0.1%) |
| Total | 1,601 | (100%) | 673 | (100%) | 928 | (100%) |

**Table 1: The cloud-based email filtering services considered in this study and their prevalence in our data.**

domains: we show the distributions of the filtering services and email providers the domains use, and examine the extent to which the domains are misconfigured and allow their filtering service to be bypassed. We end by describing various techniques we use to validate our results on a subset of the domains.

## 4.1 Filtering Services & Email Providers

We start with the results of identifying filtering services and email providers for the domains in our data set.

Table 1 shows the distribution of third-party filtering services used by the domains in our study, both overall and broken down by TLD. For each filtering service, the table shows the number of domains using the service and the percentage of all domains in each column that use the service. We note that the market is dominated by a few companies: the top five account for 90–93% of domains, with the remaining ten companies as a long tail. There is however a market variation between the two TLDs. Barracuda, for instance, is the most popular service among edu domains at 36%, but has considerably less market share in com and is ranked fourth with 4%.

Table 2 shows the number of domains that actively use each of the three email providers. Since 6% of the domains (103/1,601) actively use two providers, we include them in both counts of the providers they use (hence the total counts are slightly larger than in Table 1). In both TLDs, Exchange Online is by far the most popular provider, with a slightly higher popularity among edu domains. Gmail is the other alternative in practice, with Zoho only having four customers in our domain set.

## 4.2 Misconfigurations

Table 3 summarizes the results of inferring the integrity of the mail paths of the domains in our data set. Each cell corresponds to a filtering service and email provider in combination, and the values show the number of domains misconfigured to allow direct bypass relative to the total number of domains for that combination. For

| TLD | Exchange | Gmail | Zoho |
|---|---|---|---|
| edu | 607 (85%) | 107 (15%) | 0 (0%) |
| com | 745 (75%) | 241 (24%) | 4 (0.4%) |
| Total | 1,352 (79%) | 348 (20%) | 4 (0.2%) |

**Table 2: The number of domains inferred to use each of the three email providers in each of the TLDs. Since 6% of domains use two email providers, we include them in both counts of the providers they use (hence the total counts are slightly larger than in Table 1).**

clarity we combine the results from both TLDs and exclude the Zoho results: the misconfiguration rates are much more correlated with the combination of filtering service and email provider than which TLD the domain is in, and the four domains that use Zoho are all misconfigured and vulnerable to bypass. As with Table 2, 6% of domains use two email providers and we infer the configuration status for each provider they use and count those configurations separately in these results. Recall from Section 3.3 that evaluating bypass for domains using Exchange Online requires sending email to a valid address at that domain. For 123 domains, though, we were unable to remotely determine a valid email address. Since we could not evaluate their configuration status, we did not include those domains in these results.

Surprisingly, overall most domains misconfigure their email provider when using third-party filtering services: 80% of domains are misconfigured to allow email delivery that bypasses the filtering service. From our experience experimenting with configuring email providers to use filtering services, our conclusion is that the filtering services and the documentation they provide their customers are a significant contributing factor.

For example, domains using Cisco and TrendMicro have a higher rate of misconfiguration with Exchange Online. Cisco's [8] and TrendMicro's [60] documentation for configuring an Exchange connector omits the crucial step of restricting inbound mail to a range of gateway IP addresses. Microsoft's connector documentation describes two options for identifying email sent from third-party partner organizations — "By verifying if the sender domain matches..." and "By verifying if the IP address of the sending server matches..." [40]. When an organization chooses the latter option, the Exchange UI does not provide an option to "reject" email since the connector is only enforced during the IP matching. As a result, such a configuration can be trivially bypassed by directly delivering mail to the organization's SMTP server [57]. In contrast, Mimecast's [42] and Barracuda's [4] documentation for configuring a connector correctly describes restricting the inbound mail to gateway IP addresses as a "necessary" configuration step. We see a correspondingly lower misconfiguration rate for domains using Mimecast (66%) and Barracuda (76%).

We found similar issues with Gmail as well. For instance, most domains using Mimecast with Gmail are misconfigured (95%). Mimecast's documentation for configuring inbound email filtering for Google Workspace [43] neither explicitly instructs the administrator to restrict the inbound mail to a range of gateway IP addresses,

| Filtering Serv. | Exchange | Gmail | Total |
|---|---|---|---|
| Proofpoint | 415/541 (77%) | 152/175 (87%) | 567/716 (79%) |
| Barracuda | 186/244 (76%) | 26/27 (96%) | 212/271 (79%) |
| Mimecast | 113/171 (66%) | 69/73 (95%) | 182/244 (75%) |
| Cisco | 124/139 (89%) | 15/18 (83%) | 139/157 (89%) |
| TrendMicro | 30/30 (100%) | 10/12 (83%) | 40/42 (95%) |
| Sophos | 16/18 (89%) | 7/9 (78%) | 23/27 (85%) |
| Cloudflare | 8/8 (100%) | 10/14 (71%) | 18/22 (82%) |
| FireEye | 9/13 (69%) | 5/7 (71%) | 14/20 (70%) |
| AppRiver | 13/13 (100%) | 6/6 (100%) | 19/19 (100%) |
| ForcePoint | 11/13 (85%) | 1/1 (100%) | 12/14 (86%) |
| Fortinet | 13/14 (93%) | 1/1 (100%) | 14/15 (93%) |
| Broadcom | 10/12 (83%) | 3/3 (100%) | 13/15 (87%) |
| HornetSecurity | 2/8 (25%) | 1/1 (100%) | 3/9 (33%) |
| Solarwinds | 3/3 (100%) | – | 4/4 (100%) |
| Spamhero | 2/2 (100%) | 1/1 (100%) | 3/3 (100%) |
| Total | 955/1,229 (78%) | 307/348 (88%) | 1,263/1,578 (80%) |

**Table 3: The integrity of the mail paths of the domains in our data set. For each combination of filtering service and email provider it shows the number and percentage of misconfigured domains. Domains with two email providers are counted twice, once for each provider. For 123 domains that use Exchange Online, we could not evaluate their configuration status and exclude them from the counts in this table.**

nor highlights the risk of not doing so. In contrast, Proofpoint's documentation [47] highlights this risk and makes it a necessary step of configuration. We correspondingly observe a lower percentage of Proofpoint domains using Google to be misconfigured (87%).

However, documentation issues with filtering services are not the only explanation. Overall in our data set, domains misconfigure Gmail more often than Exchange Online: 88% of Gmail configurations allow bypass, while 78% of Exchange Online configurations do. Reviewing online blogs and forums, a number of reports mention three concerns with Gmail that reportedly interfere with normal email delivery: its "Automatically detect external IP" feature interferes with whitelisting [26]; organizations document experiences with Gmail preventing delivery from its own server IP addresses [47]; and features like Smart Banners and URL rewriting reportedly break DMARC/SPF, resulting in valid mail being labeled as spam [19]. Given these community experiences, an administrator configuring Gmail could conclude that it would be prudent not to restrict the IP ranges for incoming SMTP connections.

It is the issue of deliverability that appears to be the most common concern that could explain the prevalence of misconfigurations across the range of filtering services and email providers. The best combination of popular filtering service and email provider is Mimecast and Exchange Online. But even for the domains with that combination, it is still the case that 66% of them are at risk of a bypass attack. A repeated theme established is that of reasonable concern in ensuring successful email delivery for valid email under all scenarios. Given the importance of email communication, the goal of valid email delivery could understandably override concerns about bypass. We discuss the issue further in Section 7.

## 4.3 Validation

For a subset of the domains in our study, we used three techniques to validate the results of our inference methods. Table 4 shows the number of domains that we validated using each technique according to the filtering services used. In all cases, our validation results agree with our inference results for both the email provider for the domain and their bypass configuration status.

The first technique takes advantage of bounce messages. For some organizations that use Exchange Online, we can send email to a non-existent RCPT address and the organization will send a bounce message in response that includes the delivery path of the original message [63]. For organizations where Exchange Online is misconfigured to allow bypass, the delivery path allows us to verify that the first server to receive the message is indeed an Exchange Online server (as expected when bypassing). This technique only applies to organizations that have Exchange Online misconfigured, do not enable Directory-Based Edge Blocking, and generate bounce messages. Of the 955 organizations that misconfigure Exchange Server, 301 (32%) of them generate bounce messages, and in all cases they agree with our inference results.

The second technique uses responses from Google Groups administrative addresses to validate domains using Gmail. For example, a Google Group group@univ.edu always has a special address group+unsubscribe@univ.edu for unsubscribing. When sending email to the unsubscribe address, Google Groups responds with an error that encloses the delivery path of the original message. We can verify that the first server on the path to receive the message is a Gmail server rather than the server of the domain's cloud-filtering service. This approach, however, requires identifying Google Groups at organizations. Unfortunately, systematically searching for such addresses only discovered groups for 12 domains.

Our last technique involves personal contacts at the organization. We attempt to directly deliver email to our contact's email address. If the organization has securely configured their mail servers, then the server should reject our delivery attempt during the SMTP session. If the organization has not configured their mail servers correctly, we complete the SMTP session, send the email, and ask our contact to forward the delivered message to us. We then verify that the server we used for delivery is indeed the first server on the delivery path (rather than the server of a filtering service).

## 5 ETHICAL CONSIDERATIONS

There are two classes of ethical considerations in our work that we briefly discuss here: potential impacts to humans and potential impacts to organizations.

A straightforward approach to a study such as ours would be to simply attempt to bypass the mail filtering services used by domains under test and then leverage widely-used mail content features (e.g., embedded links to images) to establish delivery. Indeed, in discussions with our IRB office, we have been informed that sending such unsolicited emails to individuals and evaluating if they are received would not be considered human subjects research as we are not collecting information *about* the person. However, we are sensitive that such emails *still* incur a de minimus nuisance cost on recipients (i.e., reading the message and choosing what to do in response) and thus our methodology has been carefully designed

| Filtering Service | Bounce | Google Groups | Human Verifier | Total Validated |
|---|---|---|---|---|
| Proofpoint | 114 | 11 | 10 | 135 |
| Barracuda | 51 | 0 | 1 | 52 |
| Cisco | 25 | 1 | 2 | 28 |
| Others | 111 | 0 | 0 | 111 |
| Total | 301 | 12 | 13 | 326 |

Table 4: The number of domains validated using each technique and the filtering services those domains used.

to focus on purely machine-to-machine communications. However, in a minority of cases — when the domain is hosted by Microsoft's Exchange Online, is configured to filter out invalid email addresses and is incorrectly configured to allow its mail filtering service to be bypassed — we may deliver a single email to a role-oriented email (e.g. postmaster) whose identity is unknown to us.[9] In these cases, we solicit no response and perform disclosure by explaining the purpose of the study and then implications of them receiving the email.[10] Further, we conducted an interview with our institutions postmaster and they confirmed that a single message would represent "a drop in the bucket" of the mail they receive on a daily basis and would not constitute a significant differential burden.

The second issue is that our work identifies vulnerable organizations whose current mail configuration allows their mail filtering service to be bypassed. To avoid unnecessarily enabling malicious parties, we do not specifically identify vulnerable domains by name. We have disclosed these findings to the affected third-party email filtering services (as they are in the best position to help their clients fix their service deployments). Of the five filtering services who have responded thus far, three have declared the issue "out of scope" for them (since it is an issue in their customer's infrastructure and not theirs), one acknowledged the importance of the issue but that such a configuration decisions are left to the discretion of their customers, and, finally, one indicated they would consider contacting potentially misconfigured customers and updating their documentation to better document safe configuration practice.

## 6 RELATED WORK

There is a large body of prior work focused on email security and infrastructure. We highlight here the publications and reports most closely related to our study. One popular line of research examines the deployment of different email security and encryption protocols. These include efforts to characterize the real-world deployment and challenges related to STARTTLS [12, 15, 20, 33, 46], SPF [9], DKIM, DMARC [6, 7, 10, 12, 15, 22, 29, 52, 58, 62], DANE [27, 28], and PGP [55]. Separate from security pitfalls and solutions, prior work has also investigated email delivery and email service provisioning. Notably, Afergan et al. [1] examine the latency and loss aspects of email delivery and Holzbauer et al. [21] investigate protocol support in email delivery using passive DNS. Rijswijk et al [61] describe the growth of three email providers (as measured by MX records) and

---

[9]This is similar to the approach taken by Bennet et al. in their 2022 IMC paper on inferring SPF vulnerabilities [6].

[10]We also provide a link that they can use to opt out of any future messages.

Liu et al. [30] provide a large-scale measurement documenting the change of email service provisioning over time.

The prior research most related to our work focuses on the effectiveness, deployment, and adoption of cloud-based email filtering services. This literature includes Rahmad et al.'s [50] comparative study of the effectiveness of different cloud-based email filtering services, industry reports on how to defeat Proofpoint's spam filtering [45], and Fiebig et al.'s [13] and Liu et al.'s [30] measurement studies on the adoption of cloud-based email filtering services. Notably, while both of these two groups accurately identify and document the increasing use of such filtering services, they do not investigate the security implications these changes. It is these implications that motivate our work to understand the integrity of mail filtering deployments.

## 7 DISCUSSION

The problems highlighted by this paper are superficially about a set of independent failures in administrative configuration. However, the underlying reasons for these failures all stem from the larger issue of architectural inadequacy. Email, like many legacy Internet services, was designed around a simple use case that is now out of step with modern demands. In all of the examples explored in this paper, the domain owner desired to reliably route inbound mail through an *ordered set* of cloud-hosted services — first to one third-party service (the filtering service) and *then*, after filtering, to another third-party service (the mail service provider). However, this desire cannot be expressed in the existing architecture for mail delivery. Instead, the domain owner's security depends on careful coordination between its own administrators, the mail filtering service and the mail service provider to cobble together these semantics. Unsurprisingly, this ad hoc approach is rife with opportunities for failure.

First, the separation of concerns is not naturally aligned with the interests and capabilities of the parties. The mail filtering provider — the service whose very existence creates the opportunity for a bypass — is itself *incapable* of guaranteeing the integrity of mail delivery. They can forward filtered traffic on to the mail service provider, but they cannot restrict from whom that service accepts mail; only the mail service provider can do so. While the mail filtering provider is capable of implementing such a restriction, they may not provide one. Indeed, this is the norm across a range of mail service providers and in this case there is no recourse for a domain owner (except to switch providers).

However, even if such a mechanism is available, it is only effective if a domain's administrator knows of its existence and importance. We note that the configuration documentation for a number of mail filtering services make no mention of the need to configure Gmail or Exchange Online to only accept mail from their servers. Even knowing that such a mechanism exists, email administrators must then implement such inbound mail restriction correctly. This can be difficult when, in at least one significant case, the mail filter service's documentation for performing this action is inaccurate (in a way that ensures that email bypass will be possible). This knowledge issue must be overcome by each domain owner, even though the average email administrator is likely far less facile with email security than the staff at the cloud services being used.

Second, the complex federated nature of this service composition may impact (or at least be perceived to impact) mail deliverability in a way that causes domain owners to favor "open" (i.e., non-secure) implementations. For example, at least one major mail filtering service warns in its documentation that enabling inbound mail restrictions on Gmail may lead to some mail being dropped. As well, we observe a number of domains including Gmail's servers as "backup" entries in their MX records — presumably to tolerate a failure of the mail filtering provider. However, this fault-tolerant configuration can only work if Gmail is configured to accept mail from all parties (and hence, is bypassable). An added complication is that some configuration changes can take longer to propagate than others and there may be no mechanism to validate that such propagation has completed. It is due to such concerns that some mail filtering tutorials strongly advise waiting 24–48 hours between updating MX records and applying the correct configuration on the email provider's side [56]. Such indeterminacy, linked with the previously described risks of deliverability, can cause mail administrators to forego these steps to avoid potential service disruptions.

Finally, even in the best case, when everything is configured correctly, the integrity of the mail delivery path rests solely on the integrity of the IP source address — a design whose fragile security properties has long been understood [5]. Indeed, should an attacker be able to spoof the source address of a domain's mail filtering provider (e.g. communicate to Gmail servers as though one were Proofpoint) it is entirely likely that their email will avoid *all* filtering: mail filter services commonly recommend disabling spam filtering on the backend mail server to avoid interactions between disparate filtering systems (e.g., [4, 48])

Today's mail filtering ecosystem repurposes and exploits the tools available from existing mail and DNS protocols, designed long before widespread cloud deployments. The deficiencies of this approach, identified in this paper, highlight the need for a modern architecture for composing cloud services (such as mail filtering) in a way that cleanly supports strong integrity, simple configuration and transparent auditability.

## 8 CONCLUSION

Organizations have increasingly turned to cloud-based email filtering services to defend against sophisticated email threats. These services carry out their filtering function by interposing between senders and an organization's email server: incoming email messages are first processed by the filtering service before being delivered to the organization's email server. However, for this filtering function to be effective, organizations need to configure their email server to only accept email from their filtering service provider. Otherwise, malicious actors can bypass the filtering service by sending directly to the organization's email server.[11]

Using a range of com and edu domains as a case study, we empirically demonstrate that such bypasses are widely feasible 80% of the domains are configured to allow such actions. Our work highlights the stresses placed on our legacy network architecture as it is asked to solve problems — such as securely composing cloud services — that were never part of its original design.

---

[11]While attacks in wild is not the focus of this paper, we note that such bypasses have been documented [53].

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

## A HANDLING SPF EDGE CASES

In this section, we provide additional information on how we handle two edge cases: SPF records that use the `include` mechanism and SPF macros.

For SPF records that use the `include` mechanism, we expand them recursively to a maximum depth of three. At each recursion, we first check for the presence of SPF records for each provider (e.g., `include:_spf.google.com` for Gmail). If we find no match, we proceed to check if any IP addresses (`ip4` record) belongs to the suite of outbound IP address used by each provider (e.g., Exchange Online's list of outbound IP address [39]). If either of two checks succeeds, we label the domain as using the corresponding provider.

Besides the `include` mechanism, we also handle SPF records that contain macros, which are used by 8% (120/1,601) of the domains in our dataset. SPF macros provide a mechanism for dynamic SPF policies. Namely, instead specifying a list of IP addresses, it defines special sequences that are interpreted at run-time by the receiving Mail Transfer Agent (MTA). For example, the macro `%{i}` expands to the sender's IP address. If a domain's SPF record is `v=spf1 include:%{i}.spf.domain.com -all`, the receiving MTA will replace `%{i}` with the sender's IP address and then perform the SPF check by sending a DNS TXT query to `<sender's_IP_address>.spf.domain.com`. Another common macro is `%{d}`, which expands to the domain name of the sender's email address. SPF macros are designed to help avoid the ten lookup limit imposed by the DNS protocol and enable more dynamic SPF policies [51].

For a domain that has an SPF record with macros, we determine if it allows an email service provider to send on its behalf as if we received an email from that email service provider. Specifically, we expand the macros by replacing `%{i}` with an outbound IP address used by the email service provider and `%{d}` with the target domain's domain name. We then issue the DNS TXT query to the target domain. If the response indicates that the IP address used is allowed, we label the domain as using the email service provider.

