# OpenReview forum: "Unfiltered: Measuring Cloud-based Email Filtering Bypasses"
_ACM.org/TheWebConf/2024/Conference — TheWebConf24 Oral_

### Official Review · Reviewer_afse · 2023-10-28

**Novelty:** 5
**Technical Quality:** 5

**Review:**

## Summary

Some email providers on the Web use external third party email filtering providers that filter malicious contents out of emails before they reach the intended email server. The authors present an attack on such email service providers which allows to bypass this third party filtering. They further provide an analysis on the 50k most popular domains and state that most of them are vulnerable to such attacks.

## Pros

- The work is well written and easy to understand for me as a reader.

- The attack seems to be new to me and I learned a lot about email filtering providers (so thanks to the authors). As these email filtering services seem to be used a lot in practice, this attack can potentially have an impact, in case it should be scalable.

## Cons

- The practical relevance of this attack remains unclear to me. I do not see any attacker model in the paper. As stated in the paper, we have measures like DKIM and DMARC, which can authenticate whether the email was sent from the intended server. Therefore, I would assume that an attacker has to hack a legitimate email address, which is hard to scale, before this attack is even possible.

- I further see ethical issues with the work. The email filtering services that received the responsible disclosure responded that the issue lied in the customer's infrastructure. Therefore, I would expect that the vulnerable email server hostmasters should have been contacted to fix this issue. Otherwise, I assume that your work probably would allow attackers to run these attacks before giving the email service providers a chance to apply countermeasures.

## Further Notes

- Can we simply bypass email filters anyway when we use PGP or S/MIME encrypted email?

- I also see email filtering by third parties in general as a privacy and security issue, as they can read and potentially modify the whole communication of email providers. Has that issue been covered in literature somewhere?

## Conclusion

Thanks for submitting this work to WWW '24! The work and its impact could potentially be interesting for the conference. Nevertheless, I see a potentially ethical issue regarding the lack of reporting to the affected email servers. Also, the attacker models have to clearly stated in the paper to evaluate the impact of the attack. Depending on the authors' response, I can better decide on whether to accept or reject the paper.

## Post-Rebuttal

Thanks for addressing the comments in the rebuttal! According to RFC2142, online services need to have a postmaster@ email address [1], which could quickly have been generated and contacted. Also, when 20% of more than 1200 organizations would have responded to the responsible disclosure, this would mean that 240 email servers could still have been protected from such attack, which is a lot in my opinion. Therefore, I still do see ethical issues, unfortunately.

[1]: https://datatracker.ietf.org/doc/html/rfc2142

**Questions:**

- How did you handle the ethical issues regarding the vulnerable email server hostmasters?
- How relevant and scalable is the bypass attack in practice? And what is the potential impact?

**Ethics Review Description:**

The email filtering services that received the responsible disclosure responded that the issue lied in the customer's infrastructure. Therefore, I would expect that the vulnerable email server hostmasters should have been contacted to fix this issue. The paper does not state that they were contacted.

**Ethics Review Flag:**

Yes

**Reviewer Confidence:**

3: The reviewer is confident but not certain that the evaluation is correct

**Scope:**

4: The work is relevant to the Web and to the track, and is of broad interest to the community

---

### Official Review · Reviewer_oEgb · 2023-11-10

**Novelty:** 6
**Technical Quality:** 6

**Review:**

This work explores an overlook problem related with the configuration of email filtering and services. The authors conducted an empirical evaluation on approx 1,600 domain configurations and showed that 80% of these appeared to be prone to abuses. In the specific, an attacker could bypass third party email filtering providers to directly connect to the organizations' mail servers to conduct spam, phishing campaigns, etc.

The paper is well-written and easy to follow. The authors did a good job in motivating their work and presenting the methodology used to conduct the study. The results are valuable, not only in their numbers (with 80% of the organizations considered in the study affected by the problem) but also in reporting an issue that appears to be overlooked. I like the conclusion saying that the shift to the cloud introduced new challenges such as securely composing cloud services.

The ethical concerns are correctly discussed and addresses, and the identified configuration issues reported responsibly to the affected organizations.

**Questions:**

Few other comments I have are:
- Given the importance of the problem analyzed by the authors, I would have expanded the list of domains, for example by including other TLDs (on top of .edu and .com), or by doing better crawling (other than leveraging Censys/CrUX as presented in Section 3.1)
- Is there a plan to release the testing tool developed by the authors to the public to encourage large scale testing?

**Ethics Review Description:**

The ethical concerns are correctly discussed and addresses, and the identified configuration issues reported responsibly to the affected organizations.

**Ethics Review Flag:**

Yes

**Reviewer Confidence:**

3: The reviewer is confident but not certain that the evaluation is correct

**Scope:**

3: The work is somewhat relevant to the Web and to the track, and is of narrow interest to a sub-community

---

### Official Review · Reviewer_aHkU · 2023-11-23

**Novelty:** 5
**Technical Quality:** 6

**Review:**

In this paper, authors discuss how attackers could bypass third-party email filtering services employed by organizations that use popular email providers such as Google, Microsoft Exchange. Accordingly, they propose a measurement process that first infers the email service provider and email filtering service employed by a given domain (if any) and test if the filtering can be bypassed by analyzing the responses to carefully crafted messages. As a result, they conduct experiments on 673 edu domains and 928 popular domains across 15 cloud-based email filtering services. The authors find that 80% of these domains have certain misconfigurations that allows an attacker to bypass the mail delivery path via the filtering service.
This paper investigates an interesting issue of incorrect configurations in the setting up of third-party email filtering services that attackers could leverage to bypass the filtering altogether and send spam emails. Further, they leverage publicly available information such as DNS records and documentation to create a mapping between domains, email filtering services, email providers and target email addresses.

Pros:

1) Paper addresses a niche issue of bypassing email filtering
2) Investigates a broad spectrum of 15 filtering services, 3 email providers and 1601 domains
3) Provides useful insights based on their findings such as the difference observed between domains of same service and the need to redesign considering recent requirements and security. Although they do not propose any defense, this research could pave way for new solutions.

Cons:

1) As discussed in paper, their conclusions on the integrity of the services may be impacted if the domains had employed additional measures to handle such bypasses
2) More discussion required on the cases that were found to use multiple filtering services.
3) Approach is a compilation of existing techniques found in existing research and industry reports

**Questions:**

Could you provide more analysis on the overlapping cases. For instance, did your tests for integrity succeed with multiple filtering service providers, multiple email providers? In general, it would be helpful to isolate those cases in the table as well as it is hard to compare across multiple tables.

Could you provide additional information on the adoption rate of transport rules in the wild and other measures that could impact this approach?

While it poses ethical concerns if done at large scale, have you considered sending out unsolicited emails with embedded items to only few real users to validate successful delivery of the emails?

**Post-Rebuttal**
I thank the authors for their response and clarifications. I have no further questions and stand by my previous scores and review.

**Reviewer Confidence:**

3: The reviewer is confident but not certain that the evaluation is correct

**Scope:**

4: The work is relevant to the Web and to the track, and is of broad interest to the community

---

### Official Review · Reviewer_wWXT · 2023-11-24

**Novelty:** 7
**Technical Quality:** 7

**Review:**

The authors explore how an adversary could leverage that enterprises started outsourcing more and more functions to the cloud while these different functions in the cloud might not be used together in a secure fashion. The authors focused on attackers bypassing email filtering.

I enjoyed reading this paper as it was well-written, and the research was well-executed.

The authors started by developing signatures to identify mail filtering services for a selected set of domains. It would be great if the authors shared the signatures they created.

Next, the authors developed a methodology to find the mailing service providers used by these domains. The authors' method leverages the unique behaviors of these providers, the SMTP protocol, and DNS.

Finally, the authors check if bypassing the mail filtering service and sending emails directly to the mail service provider is possible. Their methodology to test if filtering can be bypassed for a domain builds on the authors' understanding of how these mail service providers behave.

The results show that most organizations allow an attacker to bypass email filtering.

My only critique is that the authors should have notified the domain owners/organizations about this vulnerability, not the filtering services.

**Questions:**

What percent of domains using Microsoft Exchange have Directory-Based Edge Blocking enabled?

What can we learn from this paper that applies to the composition of other functions outsourced to the cloud? How could we try to understand and solve such issues for the general case (for any pair of cloud services that interact)?

**Reviewer Confidence:**

3: The reviewer is confident but not certain that the evaluation is correct

**Scope:**

4: The work is relevant to the Web and to the track, and is of broad interest to the community

---

### Official Review · Reviewer_mzoA · 2023-11-28

**Novelty:** 6
**Technical Quality:** 6

**Review:**

This paper conducts a large scale of empirical study on “cloud-based email filtering bypass” vulnerability:  Although honest sending email server will send emails to the specified email server of cloud based email filtering service provider, according to the DNS entry of the receiving email server, an adversary may directly send email to the receiving email server without being filtered by the cloud service. This bypass is possible, as long as the receiving email server is not configured to reject other incoming IP addresses rather than the IP of the
cloud-based email filter service provider.

The paper also gave methods to detect which email server employs cloud-based email filtering service, and probe if the IP address based whitelist filtering is turned on in the email server.

The topic is very interesting and important. The paper is easy to follow. The methodology looks reasonable.

A minor issue is:  Is this paper the first one that discovered this vulnerability?

**Questions:**

Is this paper the first one that discovered this vulnerability?  This is not clear in the abstract and introduction.

**Ethics Review Description:**

N.A.

**Reviewer Confidence:**

3: The reviewer is confident but not certain that the evaluation is correct

**Scope:**

4: The work is relevant to the Web and to the track, and is of broad interest to the community

---

### Decision · Program_Chairs · 2024-01-22

**Decision:**

Accept (Oral)

**Comment:**

This paper demonstrates that third-party email filtering services used to reduce the amount of spam reaching incoming mail server can be effetively bypassed. The authors present a method to infer if such third-party filtering services have been misconfigured, and demonstrate that 80% of the organizations making use of popular filtering services are affected, in that their filtering services can be bypassed.

 The reviewers appreciated the novelty and technical quality of this study and the importance of the measurement findings. The presentation quality of the paper is also high, and the reviewers found the topic to be very interesting and important. Overall, both the novelty and technical quality scores are fairly high for all reviewers.

 At the same time, the reviewers noted a potential ethical issue that should be considered before acceptance. Namely, some reviewers recommended that the authors individually notify each individual affected domain owner, in addition to the affected email filtering service providers. The authors provided a detailed response to this request, making an argument that notifying all affected domains is difficult to do, given the large number. Also, they argue that notifying the affected service providers, as a way to reach their customers, is an acceptable practice that has been employed in several previous works.

 This Area Chair tends to side with the authors on this question, believing that it is reasonable to request that the authors notify the service providers, rather than each individual domain owner. In the Area Chair's opinion, the authors have provided good arguments to support this point of view.

 However, this question should also be carefully considered by the Senior Area Chairs and the PC Chairs, before making a final decision. One potential solution could be to introduce a condition for acceptance: namely, request that the authors attempt to contact all affected parties before publication, for instance by using heuristics to identify the appropriate contact information for abuse reports at each affected domain.

 ---